# Role of the Guanidinium Groups in Ligand–Receptor Binding of Arginine-Containing Short Peptides to the Slow Sodium Channel: Quantitative Approach to Drug Design of Peptide Analgesics

**DOI:** 10.3390/ijms231810640

**Published:** 2022-09-13

**Authors:** Vera B. Plakhova, Dmitriy M. Samosvat, Georgy G. Zegrya, Valentina A. Penniyaynen, Arina D. Kalinina, Ma Ke, Svetlana A. Podzorova, Boris V. Krylov, Ilya V. Rogachevskii

**Affiliations:** 1Pavlov Institute of Physiology of the Russian Academy of Sciences, 199034 Saint Petersburg, Russia; 2Ioffe Institute of the Russian Academy of Sciences, 194021 Saint Petersburg, Russia; 3Department of Pain Management, Xinhua Hospital, Shanghai Jiaotong University School of Medicine, Shanghai 200240, China

**Keywords:** arginine-containing peptides, Na_V_1.8 channel, patch-clamp method, organotypic cell culture method, conformational analysis, nociception, analgesics

## Abstract

Several arginine-containing short peptides have been shown by the patch-clamp method to effectively modulate the Na_V_1.8 channel activation gating system, which makes them promising candidates for the role of a novel analgesic medicinal substance. As demonstrated by the organotypic tissue culture method, all active and inactive peptides studied do not trigger the downstream signaling cascades controlling neurite outgrowth and should not be expected to evoke adverse side effects on the tissue level upon their medicinal administration. The conformational analysis of Ac-RAR-NH_2_, Ac-RER-NH_2_, Ac-RAAR-NH_2_, Ac-REAR-NH_2_, Ac-RERR-NH_2_, Ac-REAAR-NH_2_, Ac-PRERRA-NH_2_, and Ac-PRARRA-NH_2_ has made it possible to find the structural parameter, the value of which is correlated with the target physiological effect of arginine-containing short peptides. The distances between the positively charged guanidinium groups of the arginine side chains involved in intermolecular ligand–receptor ion–ion bonds between the attacking peptide molecules and the Na_V_1.8 channel molecule should fall within a certain range, the lower threshold of which is estimated to be around 9 Å. The distance values have been calculated to be below 9 Å in the inactive peptide molecules, except for Ac-RER-NH_2_, and in the range of 9–12 Å in the active peptide molecules.

## 1. Introduction

The modulating effects of arginine-containing short peptides on the Na_V_1.8 channel in the primary sensory neuron have aroused our recent interest [1,2,3]. Slow sodium Na_V_1.8 channels are responsible for nociceptive signal coding, and modulation of their functioning can evoke an antinociceptive effect on the organismal level [4,5]. Several agents (Ac-RRR-NH_2_, H-RRR-OH, Ac-RERR-NH_2_) have been shown to statistically significantly decrease the effective charge of the Na_V_1.8 channel activation gating system at 100 nM, while the Ac-RER-NH_2_ tripeptide was active at 1 μM [1,2,3]. On the other hand, free arginine molecules, the Ac-RR-NH_2_ dipeptide, and the Ac-REAR-NH_2_ tetrapeptide did not exhibit such an effect at 1 μM [1,2]. It has been unambiguously demonstrated that two positively charged guanidinium groups of the R^1^ and R^3^ arginine side chains are necessarily required for the effective binding of the Ac-RRR-NH_2_ and H-RRR-OH tripeptides to the suggested molecular target, the Na_V_1.8 channel [3]. These functional groups are located at the ends of relatively long and flexible side chains of the arginine residues, which makes it reasonable to assume that the distances between the guanidinium groups responsible for the ligand–receptor binding of arginine-containing peptides should fall within a certain range to provide the ligand–receptor complementarity. 

The number of arginine-containing short peptides studied up to the moment is rather limited for the purpose of clarifying the structural features of the peptides in more detail than has already been discussed above. We hypothesize that if the distance between the guanidinium groups responsible for ligand–receptor binding is less than the characteristic value of 9 Å, this peptide should not be able to modulate the Na_V_1.8 channel activation gating system. In an effort to estimate the upper threshold of the distance range within which the peptides remain active, we have designed the Ac-RAR-NH_2_ tripeptide, the Ac-RAAR-NH_2_ tetrapeptide, and the Ac-REAAR-NH_2_ pentapeptide, where the distances between the guanidinium groups might be expected to exceed this threshold. Two hexapeptides, Ac-PRERRA-NH_2_ and Ac-PRARRA-NH_2_, were designed to investigate the possible role of a rigid proline residue that might stabilize the peptide conformations, with the distances between the guanidinium groups maintained within the desired range.

It is worth noting that a set of only four amino acid residues, arginine (R), alanine (A), glutamic acid (E), and proline (P), was intentionally used to design the aforementioned and currently investigated peptides. Their terminal functional groups were made electrically neutral at physiological pH with N-terminal acylation and C-terminal amidation to eliminate the unnecessary charged moieties. Pharmacologically, this should protect the peptides from possible cleavage caused by exopeptidases during delivery to their molecular target, located in the peripheral nervous system. Structurally, the integration of additional centers of charge into the peptide molecule would strongly increment the number of different patterns of intramolecular electrostatic interactions. As a considerable consequence, the ensemble-averaged values of structural parameters might lose statistical significance due to high data dispersion. Given these limitations, the peptide conformations are expected to be mainly controlled by electrostatic repulsion between the guanidinium groups, which are always positively charged even in the bulk of the protein [6], and their attraction to the negatively charged glutamic acid carboxyl group. The prolines were suggested for the role of a rigid and sterically constrained spacer, while the alanine and glutamic acid residues were used to introduce the amino acid with non-bulky neutral or small and flexible negatively charged side chain, respectively, in an attempt to gain control over the distances between guanidinium groups without significant changes in the molecular volume when designing new arginine-containing short peptide molecules. 

The amount of experimental data allows us to suggest the existence of a common peptide binding mechanism, which might result in the discovery of a novel class of analgesic medicinal substances. The methodological approach recently applied for the Ac-RRR-NH_2_ and H-RRR-OH tripeptides [3] was also implemented herein to elucidate the ligand–receptor binding mechanism of arginine-containing short peptides to the Na_V_1.8 channel molecule. The methodology combines the patch-clamp method, the organotypic tissue culture method, and conformational analysis. Such an approach makes it possible to investigate, within a single study, the target physiological effect of the attacking molecule and its possible adverse side effects on the nerve tissue, as well as to obtain the necessary structural information. 

## 2. Results

### 2.1. The Patch-Clamp Method

Families of Na_V_1.8 sodium currents recorded in the control experiment and after the extracellular application of the Ac-PRERRA-NH_2_ and Ac-PRARRA-NH_2_ hexapeptides at 100 nM are shown in Figure 1a,b. It can be clearly seen that the amplitude values of the currents decrease in the course of the experiment, which results from the rundown effect inherent to the patch-clamp method [7]. The regular protocol [8,9] was applied to obtain normalized peak current–voltage characteristics of the sodium currents (Figure 1c,d). 

These characteristics were analyzed further to quantify the effective charge transfer (Z_eff_) carried by the Na_V_1.8 channel activation gating system. The Z_eff_ value is a quantitative measure of the Na_V_1.8 channel voltage sensitivity, a parameter of great physiological importance [4]. This parameter was introduced by Almers as the gating charge and was measured by the limiting logarithmic potential sensitivity function [10]. To evaluate it, the dependences of the chord conductance on the transmembrane potential difference G_Na_(E) were plotted based on experimental recordings. The changes in the steepness of the G_Na_(E) initial branch resulting from the application of Ac-PRERRA-NH_2_ and Ac-PRARRA-NH_2_ are more readily observed when normalized G_Na_^norm^(E) functions are constructed (Figure 2a,b). As a rule, the Boltzmann distribution is used to describe these functions. However, in the present case, it cannot be based on parameters that have a simple physical meaning. A fundamentally different approach first proposed by the creators of the membrane ionic theory [11] and then modified and elaborated by Almers [10] is, therefore, applied, which allowed us to construct Almers’ logarithmic voltage sensitivity function, L(E). 

The tangents of the slopes of the asymptotes passing through the very first points of the L(E) function determine the limiting logarithmic sensitivity of the Na_V_1.8 channel to transmembrane potential change [4,10]. The Z_eff_ value decreased from 6.7 electron charge units in the control experiment to 4.7 electron charge units after the application of Ac-PRERRA-NH_2_ (Figure 2c), and from 6.6 to 4.8 after the application of Ac-PRARRA-NH_2_ (Figure 2d). Three other peptides, Ac-RAR-NH_2_, Ac-RAAR-NH_2_, and Ac-REAAR-NH_2_, do not modulate the Na_V_1.8 channel voltage sensitivity. The effects on Z_eff_ of all short peptides studied herein are summarized in Figure 3. 

Standard statistical methods cannot be used in the framework of this approach because statistical averaging at an early stage of data processing completely hides the component that is the goal of our research. Therefore, to obtain the average Z_eff_ value, it is first required to evaluate this parameter in every single experiment. Statistical averaging was carried out over the entire ensemble of investigated cells. In the framework of the Hodgkin–Huxley formalism, the effective charge transferred by the Na_V_1.8 channel activation gating system is a conservative parameter in control conditions. According to the model-independent estimate provided by the creators of the membrane ionic theory, the control Z_eff_ value is equal to 6 electron charge units [11], which correlates with the control Z_eff_ value obtained herein for the Na_V_1.8 channel (Figure 3). 

These experimental data, together with our previous results [1,2,3], make it possible to form the basis for elucidating the very delicate molecular mechanisms of the ligand–receptor binding of arginine-containing short peptides. To understand the mechanisms in further detail, it is necessary to find a correlation between the physiological effect of the peptides and the structural data obtained by conformational analysis, which is the objective of the current study.

### 2.2. Organotypic Tissue Culture

The administration of Ac-RAR-NH_2_, Ac-RAAR-NH_2_, Ac-REAAR-NH_2_, Ac-PRARRA-NH_2_, and Ac-PRERRA-NH_2_ at 100 nM into the culturing medium did not affect the DRG neurite growth significantly (Figure 4). The area index (AI) values of the experimental explants are close to the control, which indicates that the investigated peptides do not modulate DRG neurite growth. This correlates with the results obtained earlier for Ac-RRR-NH_2_, H-RRR-OH, Ac-REAR-NH_2_, and Ac-RERR-NH_2_ [2,3] and strongly suggests that the peptides do not activate the downstream signaling cascades controlling neurite outgrowth. Otherwise, undesirable adverse side effects might be manifested upon their medicinal administration.

### 2.3. Conformational Analysis

The lowest energy conformations of Ac-RAR-NH_2_, Ac-RAAR-NH_2_, Ac-REAR-NH_2_, Ac-REAAR-NH_2_, Ac-RER-NH_2_, Ac-RERR-NH_2_, Ac-PRARRA-NH_2_, and Ac-PRERRA-NH_2_ obtained in the framework of the chosen methodology are shown in Figure 5. The Ac-RER-NH_2_, Ac-REAR-NH_2_, Ac-REAAR-NH_2_, Ac-RERR-NH_2_, and Ac-PRERRA-NH_2_ peptides were considered in two forms, where the carboxyl group of the glutamic acid side chain was either deprotonated and charged negatively or it was protonated and electrically neutral. The carboxyl group is definitely charged negatively at the dielectric constant ε = 80, which corresponds to the aqueous solution. An ε value of 10 was chosen to model the dielectric properties of the surrounding milieu at the moment of the ligand–receptor binding of the peptides to the Na_V_1.8 channel, and the protonation state of the carboxyl group is not a priori known in this case.

The ε value is 6–7 in the protein interior and can reach 20–30 at the protein–water interface [12]. Comparing the results of calculations with two different ε values makes it possible to gain some insight into the conformational changes that may occur within the attacking molecule while it moves from the extracellular solution (ε = 80) toward its binding site on the Na_V_1.8 channel molecule (ε = 10). We consider ε = 10 a reasonable value to be accepted in our implicit model, which does not take into account that the Na_V_1.8 channel interior is an anisotropic milieu expressed in a heterologous system [9]. This value was also suggested as the optimal one to be applied in homogeneous models for pKa evaluation [12]. In any case, the presented data do not indicate a major effect of the ε value on the distances between the guanidinium groups (see below). 

It should be made clear that the steric and electronic structure of a single calculated peptide conformation depends on many factors, including the chosen methodology itself. Hence, the presented molecular conformations can provide only a general idea of the spatial organization of functional groups within the peptide molecule and the possible intramolecular interactions between them. To elucidate the common structural features of the lowest energy conformational ensembles of the studied peptides, it is necessary to obtain the mean values of structural parameters over the entire ensemble of peptide conformations and smaller subensembles, which allows one to perform a kind of numerical limiting process within the entire ensemble. 

Such a protocol was applied in our previous work to investigate the Ac-RRR-NH_2_ and H-RRR-OH tripeptides and demonstrate that the R^1^–R^3^ distance value decreases with the decrement of the energy cutoff that defines the amount of lowest energy conformations in a subensemble [3]. It is the only distance between guanidinium groups in both molecules that has been shown to significantly depend on the energy cutoff value, which made it possible to suggest that the R^1^ and R^3^ guanidinium groups are directly involved in intermolecular ion–ion bonds upon the ligand–receptor binding of the tripeptides. We assumed that a reliable low-energy subensemble should include at least 1% of the total count of conformations, around 1000, to avoid the inadequate sampling of conformational space. The characteristic distance between the guanidinium groups required to provide the steric and electrostatic ligand–receptor complementarity of arginine-containing short peptides has been estimated as ~9 Å [3], and this distance corresponds to the lower threshold of the distance range, within which the peptides exhibit their target physiological effect. 

The average distances between guanidinium groups calculated over the entire ensemble and smaller subensembles in the Ac-RAR-NH_2_, Ac-RAAR-NH_2_, Ac-REAR-NH_2_, Ac-REAAR-NH_2_, Ac-RER-NH_2_, Ac-RERR-NH_2_, Ac-PRARRA-NH_2_, and Ac-PRERRA-NH_2_ molecules are presented in Table 1 and Table 2. The dielectric constant ε value was not demonstrated to have a significant effect on the distances in almost all cases, so the data obtained at ε = 10 are mainly included in the tables to avoid duplication. 

The first four peptides from the list contain only two arginine residues, and all of them have no effect on Z_eff_ in the patch-clamp experiments (results herein and in [2]). Their amino acid sequences were designed to study how a number of structural factors could modulate the average distance between guanidinium groups. The following factors were considered: the charge of non-arginine side chains (neutral alanine versus negatively charged glutamic acid) and the length of the peptide backbone and the protonation state of the glutamic acid side chain carboxyl group. The distance between the guanidinium groups was expected to increase with the elongation of the peptide backbone and, additionally, be regulated by the introduction of negatively charged side chain functional groups. However, a rather counterintuitive result, presented in Table 1, was obtained. None of the factors mentioned above was influential enough to correct the general tendency that the guanidinium groups were rather attracted to each other than repulsed. The distance between them decreased sharply in all peptides that contained only two arginine residues, from the value exceeding 10 Å over the entire ensemble of conformations to the value below 8 Å in all subensembles, which included around 1000 lowest-energy conformations. This result correlates with the earlier suggestion that the distance between the guanidinium groups involved in ligand–receptor binding with the Na_V_1.8 channel should not be less than 9 Å [3]. 

Interpreting the structural data becomes less paradoxical if the possibility of π–π stacking interactions between the guanidinium groups is taken into consideration. Closely located arginine residues have been studied both in water solutions and in protein environments [13,14], and the repulsive interactions between the charged guanidinium groups, partially stabilized by the solvation effect, were shown to be strongly stabilized by the surrounding negatively charged and polar groups. In our case, the local surrounding is not explicitly included in the model because no detailed information about the molecular structure of the arginine-containing short peptide binding site on the Na_V_1.8 channel is yet available. Nevertheless, the guanidinium groups of the four inactive peptides tend to attract each other already within the framework of the implicit solvation model used in the present calculations. Moreover, the distance value is stabilized noticeably below 9 Å, irrespective of the dielectric constant, ε; the charge of the glutamic acid side chain carboxyl group; and the length of the peptide backbone (from one to three residues between the arginines). Therefore, it seems reasonable to include the third arginine residue in the amino acid sequence of short peptides because the linear topology and the even number of guanidinium groups in the molecules of Ac-RAR-NH_2_, Ac-RAAR-NH_2_, Ac-REAR-NH_2_, and Ac-REAAR-NH_2_ result in a decrease in the distances between these functional groups in the lowest energy subensembles as compared to the average distance values over the entire ensemble of conformations, despite the extension of the amino acid sequence.

It might be very tempting to hypothesize that the third arginine is a prerequisite for the peptide to effectively modulate the Na_V_1.8 channel activation gating system. However, the Ac-RER-NH_2_ tripeptide has been found to statistically significantly decrease Z_eff_ at 1 μM [1], which puts the idea into question. According to the results of conformational analysis, the distance between guanidinium groups in the Ac-RER-NH_2_ molecule also decreases significantly below 9 Å with the decrease in the energy cutoff value if the glutamic acid side chain carboxyl group is considered negatively charged. Therefore, the observed physiological effect of this tripeptide does not fully fit into the framework of our model based on a correlation between the structural parameters and experimental data. Unfortunately, we cannot provide an unequivocal explanation of this fact. It should be noted, however, that when the glutamic acid side chain carboxyl group is considered neutral at ε = 10, a rather unexpected increase in the distance between the guanidinium groups from ~8 to 9 Å is detected with the decrease in the energy cutoff value from 3 to 2 kcal/mol. This is a unique tendency that has not been identified for any other studied peptide irrespective of its molecular form and the dielectric constant value. The protonation state of the glutamic acid side chain in the investigated peptide molecules remains unclear at the moment of their binding to the Na_V_1.8 channel, which is modeled by ε = 10. The data presented in Table 1 and Table 2 indicate that the charge of the glutamic acid side chain carboxyl group has little influence on the distance between the guanidinium groups involved in the ligand–receptor binding of the peptides, and, therefore, on their target physiological effect. It can be thus speculated that the electrically neutral form of the carboxyl group is stabilized by its interactions with the positively charged and/or polar groups of the peptide binding site on the Na_V_1.8 channel molecule. In such a case, the distance between guanidinium groups calculated over the subensemble with an energy cutoff of 2 kcal/mol is ~9 Å, which corresponds to the lower threshold of the distance range where the arginine-containing short peptides are expected to be able to modulate the Na_V_1.8 channel activation gating system. An additional supporting argument is the fact that the effect of the Ac-RER-NH_2_ tripeptide is exhibited at 1 μM, the concentration an order of magnitude higher than those of other studied active peptides. 

When the third arginine residue is introduced into the short peptide, the topology becomes nonlinear. The guanidinium groups form a two-dimensional triangle, either equilateral or somewhat distorted (Table 2). An almost ideal equilateral triangle with a side length of about 10 Å was detected in the Ac-RRR-NH_2_ and H-RRR-OH molecules [3]. Three other short peptides were designed and studied to address the effect of the third arginine in the linear unconstrained peptide structure (Ac-RERR-NH_2_) and to elucidate the possible role of the very conformationally rigid proline residue (Ac-PRARRA-NH_2_ and Ac-PRERRA-NH_2_). We hypothesize that the proline residue might have the function of a spacer, which might fix the guanidinium groups with intramolecular noncovalent interactions and maintain the required distance between them. 

The R^1^ and R^3^ guanidinium groups seem to be directly responsible for the binding of the Ac-RERR-NH_2_ tetrapeptide, as the average R^1^–R^3^ distance value exceeds 9 Å in all subensembles. The R^1^–R^4^ distance tends to decrease to a value around 8 Å with the decrement of the energy cutoff. The R^3^–R^4^ distance is fairly unaffected and close to 9 Å, but these two guanidinium groups are not simultaneously required for the effective ligand–receptor binding of arginine-containing short peptides. Were this so, the Ac-RR-NH_2_ dipeptide should be expected to modulate the Na_V_1.8 channel activation gating system, which was demonstrated to be otherwise [1]. 

In the Ac-PRARRA-NH_2_ and Ac-PRERRA-NH_2_ molecules, the R^4^–R^5^ distance (the analog of the R^3^–R^4^ distance in Ac-RERR-NH_2_) also remains almost constant at around 10 Å, irrespective of the peptide structure, the dielectric constant value, and the charge of the glutamic acid side chain. The R2–R5 distance value has a tendency to decrease by 1.5–2.5 Å to a value less than 9 Å in Ac-PRARRA-NH2 and in the form of Ac-PRERRA-NH2, where the carboxyl group of the glutamic acid side chain is neutral, and to the value of 9.3 ± 3.3 Å with an energy cutoff of 6 kcal/mol in the completely charged form of Ac-PRERRA-NH2. The R^2^–R^4^ distance between the guanidinium groups responsible for the ligand–receptor binding of the hexapeptides is around 12 Å in Ac-PRERRA-NH_2_ and around 10 Å in Ac-PRARRA-NH_2_. The proline residue is involved in two intramolecular hydrogen bonds rather conserved in the lowest energy conformations, between the R^2^ guanidinium group and the carbonyl group of the acylated N-terminus and between the R^5^ guanidinium group and the carbonyl group of the P^1^–R^2^ peptide bond (Figure 5g,h). Two of the three guanidinium groups, R^2^ and R^5^, are thus docked to the proline at 8–9 Å between them. Given the R^4^–R^5^ distance value does not change much and equals 10 Å, it is the position of the third R^4^ guanidinium group that determines the R^2^–R^4^ distance and accounts for the difference between its values in the studied hexapeptides. 

Excluding Ac-RER-NH_2_, all active peptides studied herein contain the RXRR structural motif, where three positively charged guanidinium groups located in the arginine side chains form a triangle with an R^1^–R^4^ distance of ~8–9 Å and an R^3^–R^4^ distance of ~9–10 Å. The R^1^ and R^3^ guanidinium groups located at a distance of 10–12 Å from each other are directly responsible for the ligand–receptor binding of arginine-containing short peptides to the Na_V_1.8 channel molecule. This correlates rather well with the earlier results for Ac-RRR-NH_2_ and H-RRR-OH, the simplest peptides containing three arginine residues, where all distances between the guanidinium groups are quite close to 10 Å [3]. It should be stressed that a decrease in the distance between the active guanidinium groups by only ~3 Å resulted in a complete loss of the peptide’s ability to modulate the effective charge of the Na_V_1.8 channel activation gating system. The data obtained indicate that the delicate structural tuning of the attacking molecules for effective ligand–receptor binding already takes place at the atomic level.

## 3. Discussion

The medicinal treatment of chronic pain in various etiologies usually requires the use of opiates and/or opioids that evoke adverse side effects at the organismal level and are highly addictive. For this reason, the world is experiencing an opioid crisis, which is one of the worst public health crises in history [15]. The fight against it forces us to look for approaches to create fundamentally new, effective, and safe drugs that can replace opiates and opioids in clinical practice. 

In our opinion, a prospective approach to help solve this challenging problem is to modulate the functional activity of slow sodium Na_V_1.8 channels encoding nociceptive information. The Na_V_1.8 channels are considered markers of nociceptive neurons, according to recent data [16]. An increase in their functional activity leads to an increase in the frequency of the impulse firing of nociceptors. It is the high-frequency component of impulse firing that carries information about the pain sensation to the CNS [4,17]. When pain as a sensation loses its informational and protective function and becomes chronic, this pathology can be corrected only via drug administration. Regretfully, there are no safe and effective analgesics that can replace opiates in the arsenal of clinical medicine. 

The nonlinear properties of gating machinery play a crucial role in impulse firing modulation. Such a delicate mechanism of gating system behavior in a classical sodium channel (Na_V_1.1 according to modern classification) was observed earlier in [18]. This mechanism, reflected in the weak nonlinearities of the inactivation process, provides the basis for the spike frequency adaptation phenomenon, i.e., evokes a decrease in the sensory neuron impulse firing frequency [19]. The Na_V_1.8 channel activation gating system can also be finely and directionally modulated by another mechanism decreasing the frequency of impulse firing, which was discovered by us earlier due to the application of the Almers method [4]. It is known that the agents reducing nociceptive neuron impulse firing should be considered candidates for the role of analgesic medicinal substances [4,20,21]. We hypothesize that there is an entire class of arginine-containing short peptides modulating the Na_V_1.8 channel activation gating system behavior. 

It should be noted that we implemented a quantitative approach, based on the experimental investigation of the effect of the attacking molecules on the molecular targets in identified living nerve cells, rather than cultured cell lines. This methodology makes it possible to demonstrate, in a living nociceptive neuron, whether the ligands are effectively recognized and bound to their molecular target. In the present case, it is the Na_V_1.8 channel, the functional activity of which determines the mechanism of nociceptive information coding. 

The principal result of the current work is the elucidation of the generalized ligand–receptor binding mechanism of arginine-containing short peptides with the Na_V_1.8 channel, an important member of the sodium channel superfamily. We unambiguously demonstrated earlier that the short tripeptides exclusively containing arginine amino acid residues significantly modulate the Na_V_1.8 channel’s functional activity [3]. However, the question remains open regarding the possible effects of short peptides with heterogeneous compositions, i.e., peptides that contain amino acid residues other than arginine. 

This manuscript summarizes the experimental and theoretical data obtained on eight peptides (Ac-RAR-NH_2_, Ac-RAAR-NH_2_, Ac-REAR-NH_2_, Ac-REAAR-NH_2_, Ac-RER-NH_2_, Ac-RERR-NH_2_, Ac-PRARRA-NH_2_, and Ac-PRERRA-NH_2_) using a number of methods (patch-clamp method, organotypic tissue culture, confocal microscopy, and conformational analysis). Only three of these peptides have been experimentally investigated before [1,2], and none have been the subject of conformational analysis by the protocol recently applied for the Ac-RRR-NH_2_ and H-RRR-OH tripeptides [3]. Prior results suggest the existence of a specific distance range between the guanidinium groups of the arginine side chains responsible for peptides binding to the Na_V_1.8 channel and estimate the lower threshold of this range as ~9 Å [3]. However, experimental data were not sufficient until now to validate this estimate, and our suggestion relied upon rather phenomenological knowledge. We designed and studied five new arginine-containing short peptides (Ac-RAR-NH_2_, Ac-RAAR-NH_2_, Ac-REAAR-NH_2_, Ac-PRARRA-NH_2_, and Ac-PRERRA-NH_2_), and the latter two were demonstrated to significantly decrease the Z_eff_ of the Na_V_1.8 channel activation system. This is an important physiological and, potentially, pharmacological result, as our previous data indicate that attacking molecules that specifically decrease the Z_eff_ value can be used as analgesic substances [5,20].

We also carried out a conformational analysis of all eight molecules in different forms and at different dielectric constants. The investigation of a wide selection of short peptides conducted herein supports our hypothesis about the crucial role of the two guanidinium groups of the arginine side chains in ligand–receptor binding. The distance between these two positively charged functional groups should fall within a relatively narrow range, the lower threshold of which was confirmed here as ~9 Å, based exclusively on the primary structure of the investigated peptides. Except for Ac-RER-NH_2_, all peptides containing only two arginine residues were inactive in the patch-clamp experiments, probably due to the attractive π–π stacking interactions between the guanidinium groups, which makes the distance between the two less than 9 Å. The odd number of guanidinium groups and their triangular topology makes the stacking less energetically favorable. The distances between the active guanidinium groups in the peptides with the three arginines fall within the range of 9–12 Å. The proline residues can play a special role in finetuning the above distances via hydrogen bond formation with the guanidinium groups, which restricts the conformational freedom of the latter.

Thus, the design strategy of arginine-containing short peptides can be based on predicting their target physiological effect as analgesic substances due to the fact that their presence can be correlated with a certain value of a structural parameter, the distance between guanidinium groups. Our theoretical considerations are verified in the patch-clamp experiments, where the Almers method is used to register the manifestations of the direct modulation of the Na_V_1.8 channel activation gating system by attacking molecules. 

The obtained results indicate that modulation of the Na_V_1.8 channel activation gating system occurs only when the structure of the attacking molecule is complementary to the Na_V_1.8 channel binding site within the angstrom level of accuracy. It is the requirement that has to be taken into account when designing fundamentally new effective analgesics based on arginine-rich short peptides. Another important design requirement, apart from the target physiological effect, is proof of the safety of medicinal substances under development. Due to the endogenous nature of short peptides, the manifestations of their toxic properties should be very rare. To take the first steps in this direction, we use a very sensitive organotypic cell culture method. Applied to nerve tissue, this method has made it possible to demonstrate that all the short peptides studied herein are safe on the tissue level. It is very important that they do not inhibit the growth of neurites, which indicates a high probability of the absence of adverse neurotoxic effects in the further protocol studies of arginine-containing short peptides as analgesic medicinal substances.

The present work substantiates our approach based on the combined application of the patch-clamp method, which involves the Almers quantitative analysis of the Na_V_1.8 channel responses, and conformational analysis, which should greatly facilitate the future design of analgesic peptides.

## 4. Materials and Methods

### 4.1. Chemicals and Reagents

All chemicals, excluding the arginine-containing short peptides, were purchased from Sigma (Sigma-Aldrich, St. Louis, MO, USA). The peptides were custom synthesized using classic peptide synthesis at the Verta Research and Production Company (St. Petersburg, Russia) using reagents from Sigma (Sigma-Aldrich, St. Louis, MO, USA) and Iris Biotech GmbH (Marktredwitz, Germany) and characterized with high-performance liquid chromatography (purity of more than 95%) and mass spectrometry.

### 4.2. Patch-Clamp Method 

#### 4.2.1. Quantitative Evaluation of Z_eff_

The patch-clamp method was applied to quantitively describe the effects of the attacking molecules on the Na_V_1.8 channel activation gating system by registering the Na_V_1.8 channel peak current–voltage characteristics before and after ligand–receptor binding. The method accuracy strongly depends on the series resistance, Rs [22]. Our prior data [4] indicate that if the Rs value is less than 3 MOhm, the effective charge transferred by the activation gating system of the Na_V_1.8 channel during its opening (Z_eff_, measured in electron charge units, e_0_) can be evaluated using the Almers [10] logarithmic potential sensitivity method [9] (see below). Microelectrodes with relatively large tip diameters were used in the experiments, which allowed us to maintain Rs practically constantly and below 3 MOhm. Another significant factor that might affect the accuracy of measurements is the rundown effect, manifested in a decrease in the sodium current amplitude. This effect is caused by the slow substitution of the intracellular solution with the solution inside the electrode, which results in a positive shift of the inactivation characteristics along the voltage axis and in a slow decrease in the Na_V_1.8 channel density in the neuron membrane [7,23]. It should be stressed that the Na_V_1.8 channel is exceptionally well suited for the behavior of its activation gating system to be described by the construction of the limiting conductivity function using the Almers method. The accuracy of Z_eff_ evaluation in this channel is not influenced either by the behavior of the inactivation system or the decrease in channel density due to the slow kinetics of both processes. However, the Almers method cannot be applied to the classical sodium Na_V_1.1 channel because its fast inactivation process strongly increases the error of Z_eff_ evaluation [4]. 

Unique structural and kinetic features of the Na_V_1.8 channel make it possible to register weak nonlinearities manifested only at the most negative values of the membrane potential, E. In this case, the application of the Almers method allowed us to obtain the Z_eff_ values by investigating exemplary neurons. A more detailed discourse on our patch-clamp experiment methodology was presented in recent publications [3,4].

#### 4.2.2. Dissociated Cell Culture

Dissociated sensory neurons obtained with the short-term cell culture technique were used in the experiments. Dorsal root ganglia (DRG) were isolated from the L5-S1 region of the spinal cord of newborn Wistar rats and placed in Hank’s solution. Enzymatic treatment was carried out for 2 to 5 min at 37 °C [24] using a solution composed of 1 mL Hank’s solution, 1 mL Eagle’s medium, 2 mg/mL type 1A collagenase, 1 mg/mL pronase E, and 1 mM HEPES Na, pH = 7.4. Washing and subsequent culturing were performed in Eagle’s medium with the addition of fetal bovine serum (FBS, 10 %), glucose (0.6 %), gentamicin (40 U/mL), and glutamine (2 mM). Mechanical dissociation by pipetting was carried out to obtain isolated neurons, which were then cultured in collagen-coated 40-mm Petri dishes. After 1–2 h of culturing, visually unimpaired cells were chosen for further experiments. 

#### 4.2.3. Experimental Solutions 

The following solutions were used to investigate the slow sodium Na_V_1.8 currents. The extracellular solution: 70 mM choline chloride, 65 mM NaCl, 10 mM HEPES Na, 2 mM CaCl_2_, 2 mM MgCl_2_, and 0.1 µM tetrodotoxin, pH = 7.4. The intracellular solution: 100 mM CsF, 40 mM CsCl, 10 mM NaCl, 10 mM HEPES Na, and 2 mM MgCl_2_, pH = 7.2. The pH values were adjusted with HCl. The experiments were terminated when the responses of other slower, tetrodotoxin-resistant sodium channels were visually detected in recordings of ionic currents. Single neurons were put into the experimental bath (volume 200 μL) using a micropipette. The external solution in the bath was refreshed using passive flow under gravity. The control sodium currents were recorded 10 min after a giga-ohm seal between the tip of the microelectrode and the neuron membrane formed. After that, the studied agent was added, and the sodium currents were recorded once again 15 min after the agent application. 

#### 4.2.4. Hardware and Software 

The “whole-cell recording” configuration of the patch-clamp method was implemented with the help of a hardware–software setup that comprised a patch-clamp L/M-EPC 7 amplifier, digital–analog and analog–digital converters, and a computer with a custom software package for the automated running of experiments developed in our laboratory. Data processing was aimed at constructing the logarithmic voltage sensitivity, L(E), function using the Almers method [10] and further obtaining the value of effective charge (Z_eff_, in electric charge units) transferred by the Na_V_1.8 channel activation gating system. 

The series resistance (R_S_), which determines both the dynamic and stationary errors of the method, was evaluated automatically during the experiment [22]. It should be maintained under 3 MOhm. In this case, the stationary error can be neglected if I_Na_^max^ does not exceed 1 nA. The membrane capacitance (C_m_) was also evaluated in the course of the experiment, which allowed us to automatically subtract the capacitive (I_C_) and leakage (I_L_) currents. The peak current–voltage characteristics were further constructed, making it possible to obtain the chord conductance (G_Na_) of the Na_V_1.8 channel and the Z_eff_ value. 

The following protocol of voltage Impulses was applied. The first impulse was equal to the resting potential of −60 mV. The step of holding potential, usually equal to −110 mV, was generated after. A set of sequential test impulses of 50 ms duration with an increment of 5 mV was used further to record the family of sodium currents in the voltage range from −60 to 45 mV. Registration of the amplitude (peak) values of the currents generated in response to each voltage step allowed us to build the peak current–voltage curve.

The DRG neuron membrane contained another slow sodium channel subtype, Na_V_1.9, which produces a tetrodotoxin-resistant current with wide overlap between activation and steady-state inactivation, and appears to modulate resting potential and to amplify small depolarizations [25]. This overlap makes it impossible to apply the Almers method for the quantitative evaluation of Z_eff_, so the method applicability was tested in every single experiment. Only the neurons that demonstrated the electrophysiological behavior described in detail earlier [4], i.e., the position of the current–voltage function extremum, E ≈ 0 (Figure 1c,d), were used to study the effects of the attacking molecules.

The Z_eff_ values were estimated using the limiting slope procedure [10]. The ratio of the number of open Na_V_1.8 channels (N_o_) to the number of closed channels (N_c_) is calculated as: N_o_/N_c_ = G_Na_(E)/[G_Na_^max^ − G_Na_(E)],
where G_Na_^max^ and G_Na_(E) are the maximal value and the voltage dependence of the chord conductance, respectively. G_Na_(E) can be obtained in patch-clamp experiments as: G_Na_(E) = I_peak_(E)/(E − E_Na_),
where I_peak_ is the amplitude value of the sodium current and E_Na_ is the reversal potential for sodium ions. G_Na_(E) is a monotonous function approaching its maximum G_Na_^max^ at positive E. The limiting slope procedure can be applied, according to the Almers theory, as: $$\begin{array}{ccc} \lim\left(N_{o}/N_{c}\right)= & \lim\left\{G_{\mathrm{Na}}\left(E\right)/\left[G_{\mathrm{Na}}{}^{\max}-G_{\mathrm{Na}}\left(E\right)\right]\right\}\to & C\cdot \exp\left[\left(Z_{\mathrm{eff}\;}e_{0}E\right)/\left(\mathrm{kT}\right)\right]\\ E\to -\infty & E\to -\infty & E\to -\infty \end{array}$$
where k is the Boltzmann constant, T is the absolute temperature, C is a constant, and e_0_ is the electron charge. 

The Almers method has the following practical application. When the membrane potential, E, approaches minus infinity (E → −∞), Z_eff_ can be estimated from the slope of the asymptote passing through the first experimental points determined by the very negative values of E, since Boltzmann’s principle is applicable at these potentials. The Almers limiting slope procedure makes it possible to experimentally evaluate Z_eff_ by constructing the voltage dependence of the logarithmic voltage sensitivity function, L(E): L(E) = ln (G_Na_(E)/(G_Na_^max^ − G_Na_(E)).

The asymptote passing through the very first points of the L(E) function obtained at the most negative E allows us to calculate Z_eff_, which is linearly proportional to the tangent of the asymptote slope.

### 4.3. Organotypic Nerve Tissue Culture Method

DRG explant cultures used in the experiments were obtained from 10–12 days old White Leghorn chick embryos, as described previously [3,20]. Briefly, DRG explants were placed in 40 mm Petri dishes coated with collagen and cultured for 3 days at 37 °C and 5% CO_2_ in a medium consisting of Hank’s solution (45%); Eagle’s minimal essential medium (40%); and FBS (10%) supplemented with insulin (0.5 U/mL), glucose (0.6%), L-glutamine (2 μM), and gentamicin (100 U/mL). Explants cultured in the culturing medium only served as the control (Figure 6). The neurite growth was assessed by the morphometric method. The area index (AI) was calculated as the ratio of the peripheral growth zone area to the central zone area. DRG explants were visualized using the Axio Observer Z1 microscope (Carl Zeiss, Germany). Obtained images were analyzed with ImageJ and ZEN_2012 software. Experiments were conducted using the equipment of the Confocal Microscopy Collective Use Center at the Pavlov Institute of Physiology RAS.

### 4.4. Calculational Methods

The TINKER 8.0 program package [26] with an MMFF94 force field [27] was used to carry out the conformational analysis of Ac-RAR-NH_2_, Ac-RAAR-NH_2_, Ac-REAR-NH_2_, Ac-REAAR-NH_2_, Ac-RER-NH_2_, Ac-RERR-NH_2_, Ac-PRARRA-NH_2_, and Ac-PRERRA-NH_2_. The algorithm of low-mode conformational search (LMOD) was implemented [28]. About 100,000 single searches were conducted for each peptide structure. The GB/SA approach [29] was chosen to take solvation effects into implicit account with the dielectric constant, ε = 10.0 (models dielectric properties of the surrounding milieu upon the ligand–receptor binding of the peptides to the Na_V_1.8 channel) and ε = 80.4 (aqueous solution). The guanidinium groups of the arginine side chains were always positively charged. The carboxyl groups of the glutamic acid side chains were negatively charged at ε = 80.4, but it cannot be a priori ruled out that they are protonated and electrically neutral at ε = 10.0 at the moment of ligand–receptor binding. Therefore, all peptides containing the glutamic acid residues were considered in two molecular forms.

The measure of distance between two guanidinium groups is the distance between their central carbon atoms because the carbon atom position approximately coincides with the geometric center of the guanidinium moiety, the positive charge of which is delocalized over three nitrogen atoms. Statistical data processing was conducted using our custom C++ script over the entire ensemble of ~100,000 conformations and several subensembles, which contained all conformations with energies not exceeding a certain cutoff value relative to the global minimum. A total of 7 subensembles were created based on the cutoff values of 2, 3, 4, 4.5, 5, 6, and 7 kcal/mol.

### 4.5. Statistical Analysis

The data were analyzed with STATISTICA 10.0 (StatSoft, Inc., Tulsa, OK, USA) using the Student’s *t*-test and expressed as the mean value ± SEM. Statistical significance was set at *p* < 0.05.

## Figures and Tables

**Figure 1 ijms-23-10640-f001:**
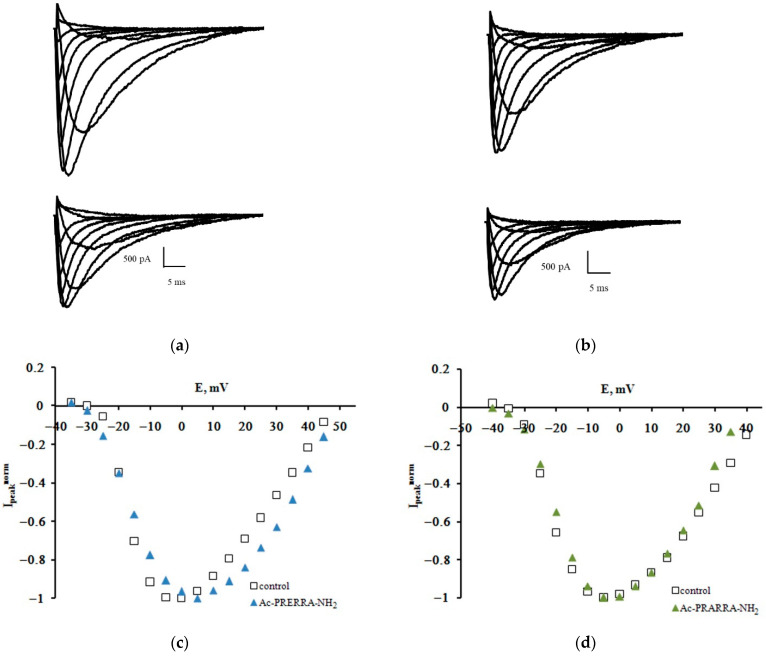
Ac-PRERRA-NH_2_ and Ac-PRARRA-NH_2_ effects on the Na_V_1.8 channel. Families of currents recorded before (top) and after (bottom) the extracellular application of Ac-PRERRA-NH_2_ (**a**) and Ac-PRARRA-NH_2_ (**b**). Normalized peak current–voltage functions of the Na_V_1.8 channel in the control experiment and after the application of Ac-PRERRA-NH_2_ (**c**) and Ac-PRARRA-NH_2_ (**d**). The test potential was changed from −60 mV to 45 mV with a step of 5 mV. The holding potential of a 300 ms duration was equal to −110 mV in all records. The leakage and capacitive currents were subtracted automatically.

**Figure 2 ijms-23-10640-f002:**
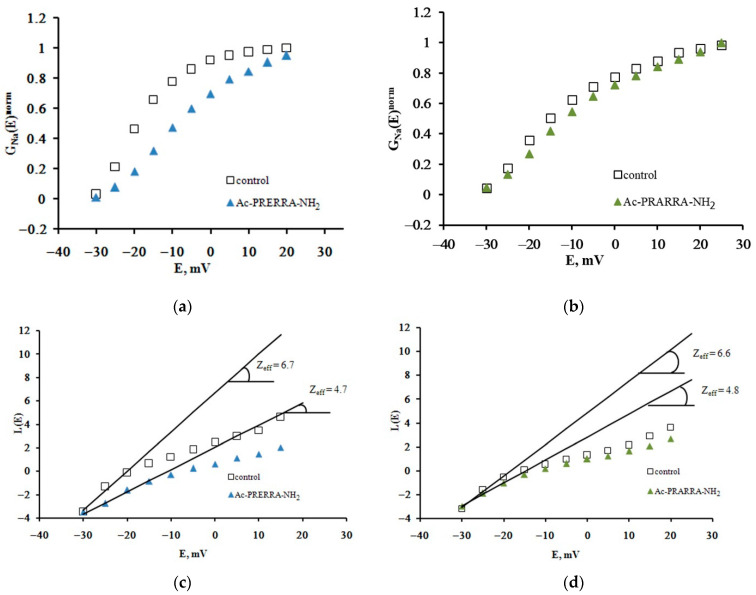
Ac-PRERRA-NH_2_ and Ac-PRARRA-NH_2_ modulate the voltage sensitivity of the Na_V_1.8 channel activation gating system. The voltage dependence of the Na_V_1.8 channel chord conductance, G_Na_(E)^norm^, in the control experiment and after the application of Ac-PRERRA-NH_2_ (**a**) and Ac-PRARRA-NH_2_ (**b**). Evaluation of Z_eff_ from the logarithmic voltage sensitivity function L(E) after the application of Ac-PRERRA-NH_2_ (**c**) and Ac-PRARRA-NH_2_ (**d**).

**Figure 3 ijms-23-10640-f003:**
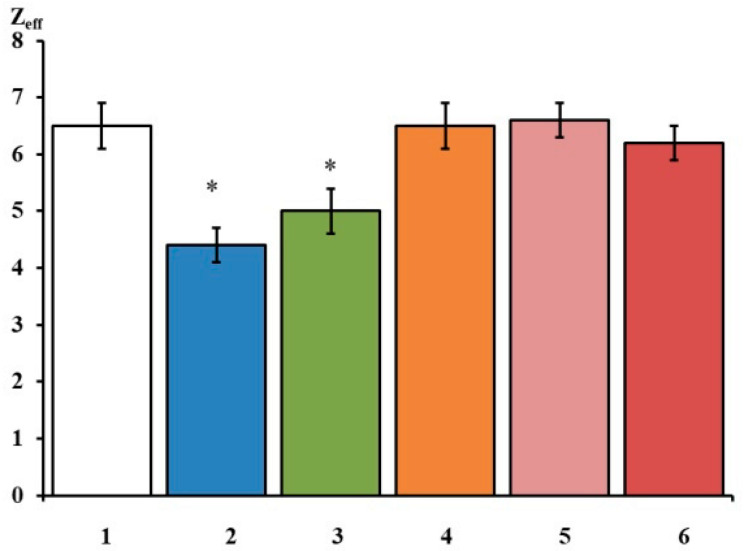
The values of effective charge of the Na_V_1.8 channel activation gating system in control conditions (1) Zeff = 6.5 ± 0.4 (n = 20) and after the application of 100 nM peptides: (2) Ac-PRERRA-NH_2_ Zeff = 4.4 ± 0.3, (3) Ac-PRARRA-NH_2_ Z_eff_ = 5.0 ± 0.4, (4) Ac-RAR-NH_2_ Z_eff_ = 6.5 ± 0.3, (5) Ac-RAAR-NH_2_ Z_eff_ = 6.6 ± 0.3, (6) Ac-REAAR-NH_2_ Z_eff_ = 6.2 ± 0.3. Data are presented as mean ± SEM. Statistically significant differences between the control and experimental values are designated with asterisks (*p* < 0.05) (n = 16–20).

**Figure 4 ijms-23-10640-f004:**
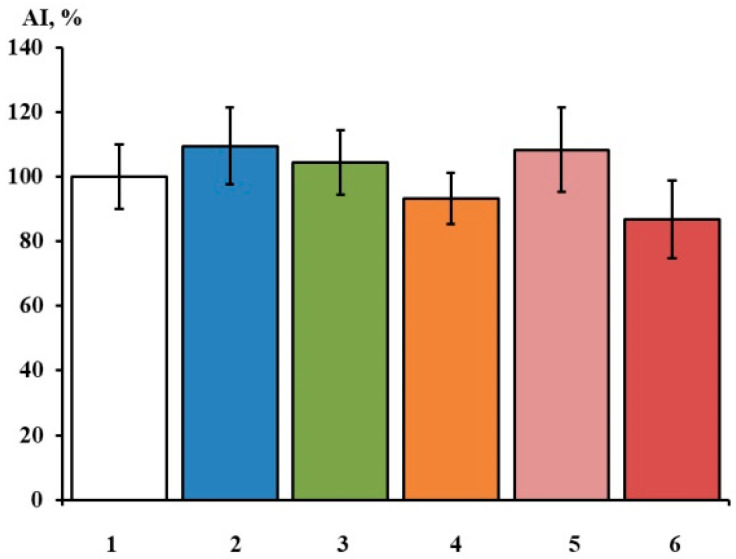
Effects of arginine-containing short peptides on neurite growth in DRG explants in control conditions (1) AI = 100 ± 10% (n = 20) and after the application of 100 nM peptides: (2) Ac-PRERRA-NH_2_ AI = 110 ± 12%, (3) Ac-PRARRA-NH_2_ AI = 105 ± 10%, (4) Ac-RAR-NH_2_ AI = 93 ± 8%, (5) Ac-RAAR-NH_2_ AI = 108 ± 13%, (6) Ac-REAAR-NH_2_ AI = 89 ± 12%. The ordinate axis—area index (AI, %). Data are presented as mean ± SEM (not significant, *p* > 0.5) (n = 16–26).

**Figure 5 ijms-23-10640-f005:**
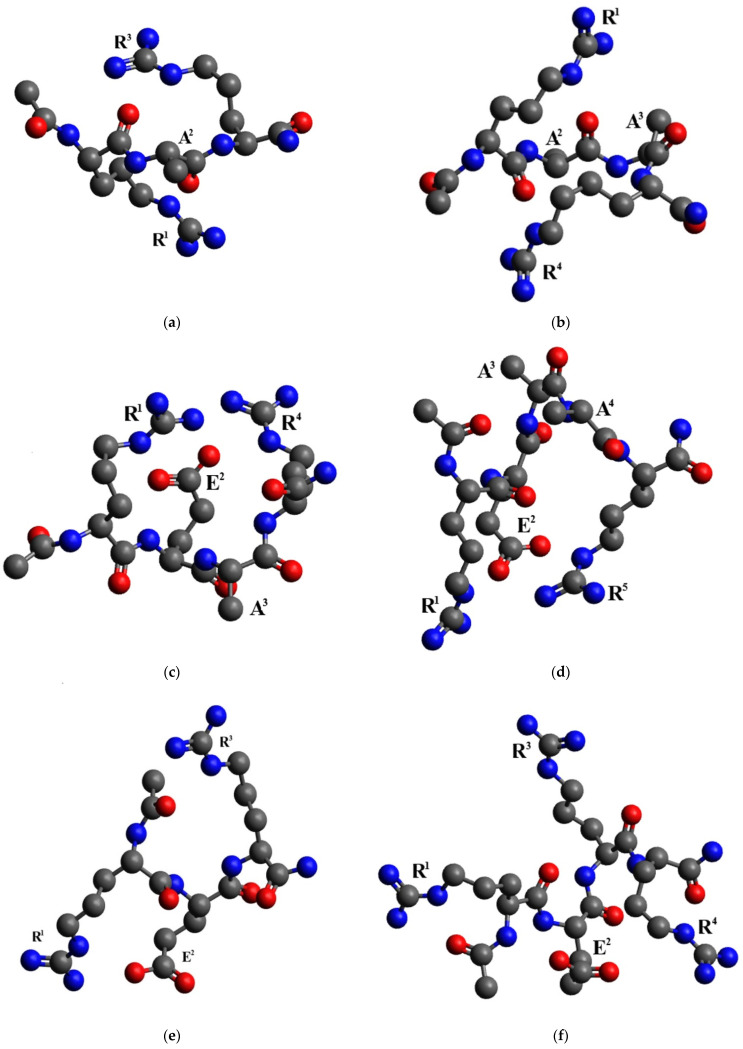
The lowest energy conformations of the peptides obtained at the dielectric constant, ε = 10. The carboxyl groups of the glutamic acid side chains were considered deprotonated and negatively charged, except for Ac-RER-NH_2_, which was considered neutral. Carbon atoms—gray spheres; oxygen atoms—red spheres; nitrogen atoms—blue spheres. Hydrogen atoms (white spheres) are not shown for clarity unless involved in highlighted hydrogen bonds. Hydrogen bonds between the proline residue and guanidinium groups of the arginine side chains in the Ac-PRARRA-NH_2_ and Ac-PRERRA-NH_2_ molecules are presented with dotted lines. The amino acid residues are enumerated. Ac-RAR-NH_2_ (**a**), Ac-RAAR-NH_2_ (**b**), Ac-REAR-NH_2_ (**c**), Ac-REAAR-NH_2_ (**d**), Ac-RER-NH_2_ (**e**), Ac-RERR-NH_2_ (**f**), Ac-PRARRA-NH_2_ (**g**), and Ac-PRERRA-NH_2_ (**h**).

**Figure 6 ijms-23-10640-f006:**
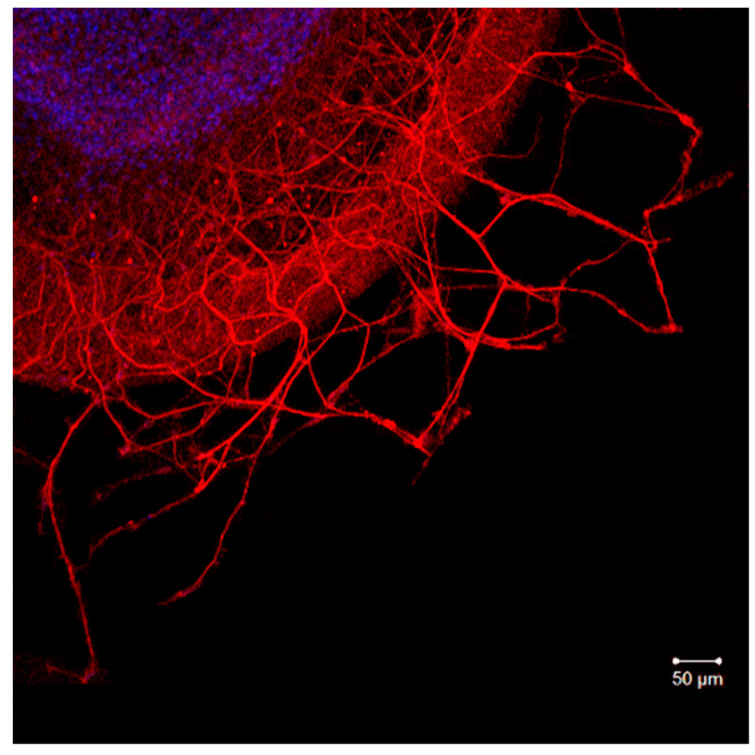
Fragment of DRG explant growth zone (third day of culturing). DRG explant is labeled with anti-neurofilament 200 antibody (red), while DAPI (blue) is used to visualize the cell’s nuclei. Scale bar 50 μm. Control.

**Table 1 ijms-23-10640-t001:** Average distances between the guanidinium groups in the molecules of arginine-containing short peptides containing two arginine residues.

Cutoff, kcal/mol	Ac-RAR-NH_2_		Ac-RAAR-NH_2_		Ac-REAR-NH_2_ ch *		Ac-REAR-NH_2_ unch *	
	ε = 10		ε = 10		ε = 10		ε = 10	
	N_conf_	Distances, Å	N_conf_	Distances, Å	N_conf_	Distances, Å	N_conf_	Distances, Å
None	102,328	R^1^–R^3^ 10.4 ± 3.6	102,412	R^1^–R^4^ 10.7 ± 4.5	103,238	R^1^–R^4^ 10.8 ± 4.2	103,022	R^1^–R^4^ 11.3 ± 4.4
7	8407	R^1^–R^3^ 8.1 ± 2.8	5903	R^1^–R^4^ 7.8 ± 2.3	2706	R^1^–R^4^ 8.1 ± 2.7	10,442	R^1^–R^4^ 8.9 ± 3.0
6	4327	R^1^–R^3^ 7.8 ± 2.5	2915	R^1^–R^4^ 7.7 ± 2.2	1073	R^1^–R^4^ 8.0 ± 2.6	6222	R^1^–R^4^ 8.5 ± 2.7
5	1882	R^1^–R^3^ 7.5 ± 2.3	1208	R^1^–R^4^ 7.8 ± 2.0	410	R^1^–R^4^ 8.1 ± 2.6	3280	R^1^–R^4^ 8.0 ± 2.4
4.5	1145	R^1^–R^3^ 7.3 ± 2.3	729	R^1^–R^4^ 7.8 ± 1.8	232	R^1^–R^4^ 8.1 ± 2.6	2233	R^1^–R^4^ 7.9 ± 2.3
4	698	R^1^–R^3^ 7.2 ± 2.2	449	R^1^–R^4^ 7.7 ± 1.8	149	R^1^–R^4^ 7.9 ± 2.4	1440	R^1^–R^4^ 7.8 ± 2.2
3	201	R^1^–R^3^ 6.6 ± 2.1	150	R^1^–R^4^ 7.8 ± 1.6	48	R^1^–R^4^ 7.6 ± 2.4	460	R^1^–R^4^ 7.8 ± 2.1
2	26	R^1^–R^3^ 6.5 ± 1.5	33	R^1^–R^4^ 7.7 ± 1.8	11	R^1^–R^4^ 7.1 ± 1.7	106	R^1^–R^4^ 7.9 ± 2.1
**Cutoff, kcal/mol**	**Ac-RER-NH_2_ ch ***		**Ac-RER-NH_2_ ch ***		**Ac-RER-NH_2_ unch ***		**Ac-RER-NH_2_ unch ***	
	**ε = 10**		**ε = 80**		**ε = 10**		**ε = 80**	
	**N_conf_**	**Distances, Å**	**N_conf_**	**Distances, Å**	**N_conf_**	**Distances, Å**	**N_conf_**	**Distances, Å**
None	102,410	R^1^–R^3^ 10.8 ± 3.6	102,660	R^1^–R^3^ 10.9 ± 3.7	102,042	R^1^–R^3^ 11.0 ± 3.6	102,538	R^1^–R^3^ 11.0 ± 3.6
7	221	R^1^–R^3^ 7.7 ± 3.0	1534	R^1^–R^3^ 8.0 ± 2.9	6870	R^1^–R^3^ 9.1 ± 3.0	7601	R^1^–R^3^ 9.2 ± 3.0
6	81	R^1^–R^3^ 7.0 ± 2.6	645	R^1^–R^3^ 7.8 ± 2.9	3262	R^1^–R^3^ 8.8 ± 2.8	3739	R^1^–R^3^ 8.8 ± 2.8
5	25	R^1^–R^3^ 6.9 ± 2.4	241	R^1^–R^3^ 7.3 ± 2.7	1298	R^1^–R^3^ 8.3 ± 2.6	1565	R^1^–R^3^ 8.5 ± 2.6
4.5	16	R^1^–R^3^ 6.7 ± 2.2	158	R^1^–R^3^ 7.3 ± 2.7	796	R^1^–R^3^ 8.2 ± 2.6	955	R^1^–R^3^ 8.4 ± 2.6
4	9	R^1^–R^3^ 7.1 ± 2.5	95	R^1^–R^3^ 7.1 ± 2.6	474	R^1^–R^3^ 8.2 ± 2.7	576	R^1^–R^3^ 8.3 ± 2.6
3	2	R^1^–R^3^ 5.2 ± 0.1	41	R^1^–R^3^ 6.6 ± 2.5	116	R^1^–R^3^ 8.3 ± 2.7	129	R^1^–R^3^ 8.0 ± 2.8
2	1	R^1^–R^3^ 5.3 ± 0.0	19	R^1^–R^3^ 6.5 ± 2.5	17	R^1^–R^3^ 9.0 ± 2.7	24	R^1^–R^3^ 7.3 ± 2.6
**Cutoff, kcal/mol**	**Ac-REAAR-NH_2_ ch ***		**Ac-REAAR-NH_2_ ch ***			**Ac-REAAR-NH_2_ unch ***		
	**ε = 10**		**ε = 80**			**ε = 10**		
	**N_conf_**	**Distances, Å**	**N_conf_**	**Distances, Å**		**N_conf_**	**Distances, Å**	
None	101,946	R^1^–R^5^ 10.7 ± 4.4	101,848	R^1^–R^5^ 10.9 ± 4.5		101,968	R^1^–R^5^ 11.1 ± 4.7	
7	1180	R^1^–R^5^ 7.7 ± 2.4	2526	R^1^–R^5^ 8.2 ± 2.6		5106	R^1^–R^5^ 8.2 ± 2.6	
6	490	R^1^–R^5^ 7.3 ± 2.2	1119	R^1^–R^5^ 8.0 ± 2.4		2310	R^1^–R^5^ 8.0 ± 2.4	
5	185	R^1^–R^5^ 6.9 ± 2.0	484	R^1^–R^5^ 7.8 ± 2.3		877	R^1^–R^5^ 7.9 ± 2.3	
4.5	126	R^1^–R^5^ 6.7 ± 2.0	289	R^1^–R^5^ 7.7 ± 2.2		510	R^1^–R^5^ 7.8 ± 2.2	
4	70	R^1^–R^5^ 6.5 ± 2.1	186	R^1^–R^5^ 7.6 ± 2.2		287	R^1^–R^5^ 7.7 ± 2.1	
3	23	R^1^–R^5^ 6.2 ± 1.4	68	R^1^–R^5^ 7.0 ± 2.4		81	R^1^–R^5^ 8.0 ± 1.9	
2	4	R^1^–R^5^ 5.8 ± 0.7	25	R^1^–R^5^ 6.8 ± 2.4		18	R^1^–R^5^ 7.8 ± 1.2	

* The subscripts “ch” and “unch” designate the protonation state of the glutamic acid side chain carboxyl group; it is either negatively charged and deprotonated or it is uncharged and protonated, respectively.

**Table 2 ijms-23-10640-t002:** Average distances between the guanidinium groups in the molecules of arginine-containing short peptides containing three arginine residues.

Cutoff, kcal/mol	Ac-RERR-NH_2_ ch *		Ac-PRERRA-NH_2_ ch *		Ac-PRERRA-NH_2_ unch *		Ac-PRARRA-NH_2_	
	ε = 10		ε = 10		ε = 10		ε = 10	
	N_conf_	Distances, Å	N_conf_	Distances, Å	N_conf_	Distances, Å	N_conf_	Distances, Å
None	101,546	R^1^–R^3^ 9.8 ± 3.4 R^1^–R^4^ 9.2 ± 3.7 R^3^–R^4^ 9.4 ± 2.7	102,238	R^2^–R^4^ 12.9 ± 3.4 R^2^–R^5^ 10.7 ± 3.7 R^4^–R^5^ 10.3 ± 2.7	101,716	R^2^–R^4^ 12.6 ± 3.6 R^2^–R^5^ 10.6 ± 3.7 R^4^–R^5^ 10.1 ± 2.8	101,517	R^2^–R^4^ 10.0 ± 3.5 R^2^–R^5^ 10.0 ± 4.1 R^4^–R^5^ 10.7 ± 2.3
7	2354	R^1^–R^3^ 9.7 ± 3.1 R^1^–R^4^ 8.0 ± 2.8 R^3^–R^4^ 9.1 ± 2.4	3389	R^2^–R^4^ 12.4 ± 2.8 R^2^–R^5^ 9.3 ± 3.2 R^4^–R^5^ 10.1 ± 2.4	5661	R^2^–R^4^ 12.3 ± 2.9 R^2^–R^5^ 8.8 ± 3.3 R^4^–R^5^ 10.0 ± 2.4	4756	R^2^–R^4^ 10.2 ± 3.1 R^2^–R^5^ 9.0 ± 4.1 R^4^–R^5^ 10.7 ± 2.2
6	1023	R^1^–R^3^ 9.7 ± 3.1 R^1^–R^4^ 8.2 ± 2.7 R^3^–R^4^ 9.2 ± 2.3	1600	R^2^–R^4^ 12.1 ± 2.8 R^2^–R^5^ 9.3 ± 3.3 R^4^–R^5^ 10.1 ± 2.4	2885	R^2^–R^4^ 12.2 ± 2.8 R^2^–R^5^ 8.2 ± 3.3 R^4^–R^5^ 9.9 ± 2.2	2353	R^2^–R^4^ 10.0 ± 3.0 R^2^–R^5^ 8.9 ± 4.2 R^4^–R^5^ 10.6 ± 2.1
5	408	R^1^–R^3^ 9.3 ± 3.2 R^1^–R^4^ 8.2 ± 2.4 R^3^–R^4^ 9.3 ± 2.3	663	R^2^–R^4^ 11.8 ± 2.7 R^2^–R^5^ 9.2 ± 3.3 R^4^–R^5^ 10.1 ± 2.3	1242	R^2^–R^4^ 12.1 ± 2.4 R^2^–R^5^ 8.1 ± 3.4 R^4^–R^5^ 9.9 ± 2.2	1030	R^2^–R^4^ 10.1 ± 2.9 R^2^–R^5^ 8.7 ± 4.1 R^4^–R^5^ 10.6 ± 2.0
4.5	258	R^1^–R^3^ 9.1 ± 3.3 R^1^–R^4^ 8.1 ± 2.4 R^3^–R^4^ 9.4 ± 2.2	411	R^2^–R^4^ 11.6 ± 2.7 R^2^–R^5^ 9.3 ± 3.3 R^4^–R^5^ 10.1 ± 2.2	728	R^2^–R^4^ 12.1 ± 2.3 R^2^–R^5^ 8.1 ± 3.4 R^4^–R^5^ 9.9 ± 2.2	629	R^2^–R^4^ 10.0 ± 2.8 R^2^–R^5^ 8.6 ± 4.0 R^4^–R^5^ 10.5 ± 2.0
4	157	R^1^–R^3^ 8.6 ± 3.2 R^1^–R^4^ 8.2 ± 2.4 R^3^–R^4^ 9.5 ± 2.2	235	R^2^–R^4^ 11.4 ± 2.8 R^2^–R^5^ 9.5 ± 3.4 R^4^–R^5^ 10.1 ± 2.2	417	R^2^–R^4^ 12.1 ± 2.2 R^2^–R^5^ 8.0 ± 3.1 R^4^–R^5^ 10.0 ± 2.1	364	R^2^–R^4^ 10.0 ± 2.8 R^2^–R^5^ 8.3 ± 3.9 R^4^–R^5^ 10.4 ± 1.9
3	55	R^1^–R^3^ 7.6 ± 2.9 R^1^–R^4^ 8.7 ± 2.1 R^3^–R^4^ 9.9 ± 2.0	82	R^2^–R^4^ 10.7 ± 2.9 R^2^–R^5^ 10.1 ± 3.3 R^4^–R^5^ 10.1 ± 2.1	91	R^2^–R^4^ 11.7 ± 2.3 R^2^–R^5^ 7.2 ± 2.6 R^4^–R^5^ 9.7 ± 2.0	105	R^2^–R^4^ 9.7 ± 2.8 R^2^–R^5^ 8.2 ± 3.4 R^4^–R^5^ 10.3 ± 1.9
2	21	R^1^–R^3^ 6.3 ± 2.3 R^1^–R^4^ 8.3 ± 1.6 R^3^–R^4^ 9.8 ± 1.5	25	R^2^–R^4^ 9.8 ± 3.0 R^2^–R^5^ 10.3 ± 3.1 R^4^–R^5^ 9.8 ± 1.8	24	R^2^–R^4^ 12.3 ± 2.3 R^2^–R^5^ 7.9 ± 3.1 R^4^–R^5^ 9.2 ± 2.1	31	R^2^–R^4^ 8.5 ± 2.9 R^2^–R^5^ 9.2 ± 3.7 R^4^–R^5^ 10.3 ± 1.8

* The subscripts “ch” and “unch” designate the protonation state of the glutamic acid side chain carboxyl group; it is either negatively charged and deprotonated or it is uncharged and protonated, respectively.

## Data Availability

Not applicable.

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
