# Peer review of "Role of the Guanidinium Groups in Ligand–Receptor Binding of Arginine-Containing Short Peptides to the Slow Sodium Channel: Quantitative Approach to Drug Design of Peptide Analgesics"

_ijms, 2022, doi:10.3390/ijms231810640_

Round 1

Reviewer 1 Report

The authors here identify that guanidinium groups of short arginine-containing peptides are important for binding to pain-associated voltage-gated sodium channels. These peptides were shown to modulate the channels only when the guanidinium group is 9-12 A away from the sodium channels.  This study suggests a common peptide binding mechanism to sodium channels and also a potential analgesic substance. This could be interesting to many readers but in my opinion, this manuscript is not yet ready for publication and needs more clarifications.

Major comments:

1. I can see that a lot of texts are from previous papers. Please avoid that.

2. Many results and conclusions are similar to what they recently published, in other words, nothing really new is shown in the manuscript.

3. Figure 1 is confusing, please add more labels.

4. Line 100, since there is a run-down issue of the currents, how could the authors know the decreased currents are due to the run-down or peptides?

5. Line 102, I would like more explanations about Figures 1c and 1d. What is the effect of peptides on the I-V curve? Also, any mechanism behind the increased currents at positive voltages in Figure 1c.

6. Line 170, it would be much better to show the images of neurite growth as the authors did for a previous paper. (https://doi.org/10.3390/ijms23115993)

Author Response

  1. I can see that a lot of texts are from previous papers. Please avoid that.

Response: Lines 34-92, 376-474.

  1. Many results and conclusions are similar to what they recently published, in other words, nothing really new is shown in the manuscript.

Response:  Lines 17-30.

  1. Figure 1 is confusing, please add more labels.

Response: done. Figure 1, lines 107-113.

  1. Line 100, since there is a run-down issue of the currents, how could the authors know the decreased currents are due to the run-down or peptides?

Response: done. Lines 486-515.

  1. Line 102, I would like more explanations about Figures 1c and 1d. What is the effect of peptides on the I-V curve? Also, any mechanism behind the increased currents at positive voltages in Figure 1c.

Response: done. Lines 507-515.

  1. Line 170, it would be much better to show the images of neurite growth as the authors did for a previous paper. (https://doi.org/10.3390/ijms23115993)

Response: done. Figure 6.

Reviewer 2 Report

The authors examined the effects of arginine-containing short peptide on slow sodium channel by using whole-cell patch-clamp recording, cell cultures om DRG neurons and combined with conformational analysis. The results suggest that the peptides reported here might have some side effects on nervous system. I have one minor concern.

1. Regarding the recording of the sodium currents as listed in Figure 1a and 1b. The authors mentioned that there are run-down during the recording process, but the authors did mentioned how long it takes to record the first trace (before treatment as a control) and second trace (after peptide treatment) in the figure 1a and 1b. It should be described in the results or methods for this recording process.

Author Response

  1. Regarding the recording of the sodium currents as listed in Figure 1a and 1b. The authors mentioned that there are run-down during the recording process, but the authors did mentioned how long it takes to record the first trace (before treatment as a control) and second trace (after peptide treatment) in the figure 1a and 1b. It should be described in the results or methods for this recording process.

Response: done. Lines 538-541, 486-515.

Reviewer 3 Report

Please find below comments and some questions on the manuscript "Role of the Guanidinium Groups in Ligand-Receptor Binding of Arginine-Containing Short Peptides to the Slow Sodium Channel: Quantitative Approach to Drug Design of Peptide Analgesics".

P1 L17 “Several of short arginine-containing peptides investigated are shown”

“investigated” might be deleted

P1 L18 Term “activation gating device” almost never found in the literature, it should be modified to “activation gating system” or “activation gating mechanism” here and later in the text. Please, explain what do you mean using this term.

P1 L27 “bonding” should be modified to “binding” here and later in the text

P1 L 35-38 Sentence “Investigating the modulating effects of short arginine-containing peptides on the NaV1.8 channel in the primary sensory neuron, we have shown earlier that several agents (Ac-RRR-NH2, H-RRR-OH, Ac-RERR-NH2) statistically significantly decrease the effective charge of the NaV1.8 channel activation gating device when applied in the nanomolar range of concentrations [1,2].” is very difficult to understand, please rephrase.

P1 L41 Phrase “while the Ac-RRR-NH2 tripeptide did ” should be deleted since it duplicates information in the sentence “we have shown earlier that several agents (Ac-RRR-NH2, H-RRR-OH, Ac-RERR-NH2) statistically significantly decrease the effective charge of the NaV1.8 channel activation gating device when applied in the nanomolar range of concentrations [1,2]” (P1 L 35-39). In addition there is some confusion, is Ac-RRR-NH2 active at micromolar range (as referred in L41) or nanomolar range (L38)?

P1 L41 “activation gating device” might be deleted.

L 41-44 Sentences “It has been unambiguously demonstrated in our recent publication that two positively charged guanidinium groups of the arginine side chains are necessarily required for effective peptide binding to the suggested molecular target, the NaV1.8 channel activation gating device [2].” and P2 L48-51 “Conformational analysis of the Ac-RRR-NH2 and H-RRR-OH tripeptides has revealed that the R1 and R3 guanidinium groups are directly involved in intermolecular electrostatic bonding between the agents and the NaV1.8 channel molecule [2].” are duplicates of the main idea. They both separated by L 52-71 from the further discussion of arginine side chains relative position within L 76-85. I recommend unifying this statement into one fragment.

P2 L52 “It is worth noting that a very restricted set of only four amino acid residues arginine 52 (R), alanine (A), glutamic acid (E), and proline (P) was intentionally used to” should be modified to “It is worth noting that a set of only four amino acid residues: arginine  (R), alanine (A), glutamic acid (E), and proline (P), was intentionally used to”. Further, the names of a.a. must either be given in full without the one-letter code within brackets, or as one- or three-letter code.

P2 L52 “carboxy” should be modified to “carboxyl”

P2 L71-74 The sentence “Slow sodium NaV1.8 channels are responsible for nociceptive signal coding, and modulation of their functioning can evoke an antinociceptive effect at the organismal level [5].” should be placed before the Line 35.

P2 L75 “to provide enough structural information to clarify this mechanism in more detail than it has been already investigated by us and discussed above” should be modified to “to provide enough structural information to clarify interaction mechanism in more detail than it has been already investigated by us and discussed above” or rather to “to provide enough structural information to clarify peptide structural features in more detail than it has been already investigated by us and discussed above”

P2 L85 The point of phrase “physiologically productive peptide conformations” is not clear in these context. Please, rephrase.

P2 L86-87 “The amount of experimental data is sufficient to suggest the existence of a common peptide binding mechanism and, potentially, a novel class of analgesic medicinal substances”. Premature conclusion in introduction and in the manuscript overall. Studies of peptides on other ion channel subtypes and in vivo experiments are needed.

P3 L116 “These characteristics should be analyzed further” might be modified to “These characteristics were analyzed further”

P3 L117-119 “The Zeff value is a quantitative measure of the NaV1.8 channel voltage sensitivity, a parameter of great physiological importance [].” The reference should be added.

P3 L163-166 This sentence rather fit Discussion section then Results one.

P4 L131-136 The Zeff value should be presented as mean ± SEM in the text.

Is it possible to replace Zeff values in figures 2c and 2d and in the text to control value of effective charge, Zeff = 6.5 ± 0.4 given in figure 3?

In figures 1 and 2, confidence intervals and centerline should be indicated. The caption for Figures 2c and 2d should be rephrased.

Caption for Figures 3 and 4 should line up. It would be more appropriate as following “The values of effective charge of the NaV1.8 channel activation gating system in control condition (1) Zeff = 6.5 ± 0.4 (n = 20) and after the application of  100 nM peptides: (2) Ac-PRERRA-NH2 Zeff = 4.4 ± 0.3, (3) Ac-PRARRA-NH2 Zeff = 5.0 ± 0.4, … Statistically significant differences between the control and experimental values are designated with asterisks (p < 0.05) (n = 16-20).”

P5 L170 “AI values” should be modified to “area index (AI) values”

P5 L172-173 “It correlates with the results obtained earlier for Ac-RRR-NH2, H-RRR-OH, Ac-REAR-NH2, Ac-RERR-NH2 [],” The reference should be added.

P6 L201 “forms” should be replaced to “conformations”

P6 L203 The point of phrase “numerical conclusions” is not clear. Please, rephrase.

P8 L220 “charged, except for” Color the comma black.

P8 In figure 5 hydrogen atoms forming hydrogen bonds are not shown. Why are other types of interactions not shown, in particular π-π stacking interactions, which are discussed in the Conformational analysis section?

A space between R–R and distance values should be added in the tables 1 and 2.

P11 L265 “factors were to be considered” might be replaced to “factors were considered”

P12 L336 “which might dock the guanidinium groups by intramolecular noncovalent interactions and maintain the required distance between them” might be replaced to “which might fix the relative position of guanidinium groups by intramolecular noncovalent interactions and maintain the required distance between them” since the meaning of “dock” term is not clear before the detailed explanation in L355-359.

P13 L362-364 “Quite similar features were observed in the Ac-RERR-NH2 tetrapeptide with slightly different distance values [], which makes it possible to generalize the obtained data” The reference should be added.

P13 L388 “These channels are considered markers of nociceptive neurons” should be replaced to “These channels are considered to be/as markers of nociceptive neurons”

Author Response

P1 L17 “Several of short arginine-containing peptides investigated are shown”

“investigated” might be deleted

Response: done. Line 17

P1 L18 Term “activation gating device” almost never found in the literature, it should be modified to “activation gating system” or “activation gating mechanism” here and later in the text.

Response: done. Lines 18, 39, 46, 57 and further throughout the manuscript

Please, explain what do you mean using this term.

Response: done. Lines 39-40

P1 L27 “bonding” should be modified to “binding” here and later in the text

Response: done. Lines 27, 44, 58-59, 223, 451, 461

P1 L 35-38 Sentence “Investigating the modulating effects of short arginine-containing peptides on the NaV1.8 channel in the primary sensory neuron, we have shown earlier that several agents (Ac-RRR-NH2, H-RRR-OH, Ac-RERR-NH2) statistically significantly decrease the effective charge of the NaV1.8 channel activation gating device when applied in the nanomolar range of concentrations [1,2].” is very difficult to understand, please rephrase.

Response: done. Lines 34-35, 37-41

P1 L41 Phrase “while the Ac-RRR-NH2 tripeptide did ” should be deleted since it duplicates information in the sentence “we have shown earlier that several agents (Ac-RRR-NH2, H-RRR-OH, Ac-RERR-NH2) statistically significantly decrease the effective charge of the NaV1.8 channel activation gating device when applied in the nanomolar range of concentrations [1,2]” (P1 L 35-39). In addition there is some confusion, is Ac-RRR-NH2 active at micromolar range (as referred in L41) or nanomolar range (L38)?

Response: done. Line 42

P1 L41 “activation gating device” might be deleted.

Response: we have decided not to delete it

L 41-44 Sentences “It has been unambiguously demonstrated in our recent publication that two positively charged guanidinium groups of the arginine side chains are necessarily required for effective peptide binding to the suggested molecular target, the NaV1.8 channel activation gating device [2].” and P2 L48-51 “Conformational analysis of the Ac-RRR-NH2 and H-RRR-OH tripeptides has revealed that the R1 and R3 guanidinium groups are directly involved in intermolecular electrostatic bonding between the agents and the NaV1.8 channel molecule [2].” are duplicates of the main idea. They both separated by L 52-71 from the further discussion of arginine side chains relative position within L 76-85. I recommend unifying this statement into one fragment.

Response: done. Lines 42-46

P2 L52 “It is worth noting that a very restricted set of only four amino acid residues arginine 52 (R), alanine (A), glutamic acid (E), and proline (P) was intentionally used to” should be modified to “It is worth noting that a set of only four amino acid residues: arginine  (R), alanine (A), glutamic acid (E), and proline (P), was intentionally used to”. Further, the names of a.a. must either be given in full without the one-letter code within brackets, or as one- or three-letter code.

Response: done. Line 51

P2 L52 “carboxy” should be modified to “carboxyl”

Response: done. Lines 64, 181, 183, 186 and further throughout the manuscript

P2 L71-74 The sentence “Slow sodium NaV1.8 channels are responsible for nociceptive signal coding, and modulation of their functioning can evoke an antinociceptive effect at the organismal level [5].” should be placed before the Line 35.

Response: done. Lines 35-37

P2 L75 “to provide enough structural information to clarify this mechanism in more detail than it has been already investigated by us and discussed above” should be modified to “to provide enough structural information to clarify interaction mechanism in more detail than it has been already investigated by us and discussed above” or rather to “to provide enough structural information to clarify peptide structural features in more detail than it has been already investigated by us and discussed above”

Response: done. Line 71

P2 L85 The point of phrase “physiologically productive peptide conformations” is not clear in this context. Please, rephrase.

Response: done. Lines 80-82

P2 L86-87 “The amount of experimental data is sufficient to suggest the existence of a common peptide binding mechanism and, potentially, a novel class of analgesic medicinal substances”. Premature conclusion in introduction and in the manuscript overall. Studies of peptides on other ion channel subtypes and in vivo experiments are needed.

Response: done. Lines 83-84

P3 L116 “These characteristics should be analyzed further” might be modified to “These characteristics were analyzed further”

Response: done. Lines 114

P3 L117-119 “The Zeff value is a quantitative measure of the NaV1.8 channel voltage sensitivity, a parameter of great physiological importance [].” The reference should be added.

Response: done. Line 117

P3 L163-166 This sentence rather fit Discussion section then Results one.

Response: We understand the reviewer’s point. However, we have decided to keep the sentence as it was. The objective of this sentence is to emphasize once again that the correlation we are discussing further is based exclusively on the set of experimental data presented above.

P4 L131-136 The Zeff value should be presented as mean ± SEM in the text.

Response:  Lines 147-152, 510-515.

Is it possible to replace Zeff values in figures 2c and 2d and in the text to control value of effective charge, Zeff = 6.5 ± 0.4 given in figure 3?

Response:  Lines 507-515

In figures 1 and 2, confidence intervals and centerline should be indicated. The caption for Figures 2c and 2d should be rephrased.

Response:. Lines 507-515

Caption for Figures 3 and 4 should line up. It would be more appropriate as following “The values of effective charge of the NaV1.8 channel activation gating system in control condition (1) Zeff = 6.5 ± 0.4 (n = 20) and after the application of  100 nM peptides: (2) Ac-PRERRA-NH2 Zeff = 4.4 ± 0.3, (3) Ac-PRARRA-NH2 Zeff = 5.0 ± 0.4, … Statistically significant differences between the control and experimental values are designated with asterisks (p < 0.05) (n = 16-20).”

Response: done. Lines 147-152, 171-175

P5 L170 “AI values” should be modified to “area index (AI) values”

Response: done. Line 162

P5 L172-173 “It correlates with the results obtained earlier for Ac-RRR-NH2, H-RRR-OH, Ac-REAR-NH2, Ac-RERR-NH2 [],” The reference should be added.

Response: done. Line 165

P6 L201 “forms” should be replaced to “conformations”

Response: done. Line 190

P6 L203 The point of phrase “numerical conclusions” is not clear. Please, rephrase.

Response: done. Lines 192-193

P8 L220 “charged, except for” Color the comma black.

Response: done. Line 210

P8 In figure 5 hydrogen atoms forming hydrogen bonds are not shown. Why are other types of interactions not shown, in particular π-π stacking interactions, which are discussed in the Conformational analysis section?

Response: done. Figures 5gh. Hydrogen atoms involved in discussed hydrogen bonds are presented in the figures. π-π stacking interactions are not shown in the picture to avoid confusion. The tendency for attraction of the guanidinium groups due to stacking is manifested in the ensemble-averaged values of the distances between these groups. The single lowest energy conformations are given in the figures mostly to illustrate the spatial structure of the studied peptides. The attractive stacking interactions are not present in these specific conformations, which follows from the figures. However, we might highlight the possible π-π stacking interactions with the arrows if it helps to understand our point more clearly. 

A space between R–R and distance values should be added in the tables 1 and 2.

Response: done

 P11 L265 “factors were to be considered” might be replaced to “factors were considered”

Response: done. Line 265

P12 L336 “which might dock the guanidinium groups by intramolecular noncovalent interactions and maintain the required distance between them” might be replaced to “which might fix the relative position of guanidinium groups by intramolecular noncovalent interactions and maintain the required distance between them” since the meaning of “dock” term is not clear before the detailed explanation in L355-359.

Response: done. Line 335

P13 L362-364 “Quite similar features were observed in the Ac-RERR-NH2 tetrapeptide with slightly different distance values [], which makes it possible to generalize the obtained data” The reference should be added

Response: sentence deleted

P13 L388 “These channels are considered markers of nociceptive neurons” should be replaced to “These channels are considered to be/as markers of nociceptive neurons”

Response: done. Line 379

Round 2

Reviewer 1 Report

In general, they didn’t address my comments. Details are in comments 1, 2, 3, and 5. They have copied and cited a lot of work from their previous work. In addition, I don’t see much novelty and attraction for many readers. 

Author Response

Reviewer 1

In general, they didn’t address my comments. Details are in comments 1, 2, 3, and 5. They have copied and cited a lot of work from their previous work. In addition, I don’t see much novelty and attraction for many readers.

Major comments:

  1. I can see that a lot of texts are from previous papers. Please avoid that.

Around 110 newly written lines were added to the text (186-195, 401-464, 493-530, 581-591). The duplication report available upon request demonstrated the 28% similarity index after the first round of revisions, which was acceptable for our previous paper. It should be noted that many terms used in the manuscript do not have any reliable synonyms which makes duplication inevitable. Moreover, we apply standardized experimental techniques and procedures for at least a decade which makes it difficult to completely rephrase the corresponding parts of Methods and Results in every publication. We would also like to note that Discussion herein is almost free from previously used phrasing, as well as the major part of Results and Introduction. The initial part of Introduction mainly covers the earlier obtained data, hence some text borrowing.

  1. Many results and conclusions are similar to what they recently published, in other words, nothing really new is shown in the manuscript.

Done. Lines 419-457

  1. Figure 1 is confusing, please add more labels.

Could the Reviewer specify in more detail what was confusing, please? We thought that the figure might be overloaded with data and therefore removed the voltage steps protocol, which was intended to make the figure less confusing. We have not ignored the Reviewer’s remark.

Done lines 97-98, 708-709

  1. Line 102, I would like more explanations about Figures 1c and 1d. What is the effect of peptides on the I-V curve? Also, any mechanism behind the increased currents at positive voltages in Figure 1c.

The normalized “peak” current-voltage characteristics were constructed using the “classical” protocol, according to the Hodgkin-Huxley methodology ([8], see also [9]: fig.2.9). These “peak” values of currents never increased as compared to the control values in our experiments. It is clearly seen in Fig 1 a,b where the native families of currents are presented in control conditions and after the agent application.

The Reviewer has specified that his remark relates to the mechanism behind the increased currents at positive voltages in Figure 1c. There is no mechanism that increases the amplitudes of the currents. The effect of peptides results in a slight shift of the current-voltage function to the right along the voltage axis E. This effect may result from the change of the membrane surface charge after the attacking molecule has been applied. The mechanism of this effect can be described in the framework of the Gouy-Chapman model and deserves a special attention. However, it is out of consideration in the present manuscript that studies the behavior of the Nav1.8 channel activation gating system. It should be stressed that this effect does not influence the effective charge value (Zeff), which is well illustrated in Fig. 1 (see also:[4]).

Does the Reviewer think that the above considerations should be included in the text?

Reviewer 3 Report

Please, find below additional comments after the first round of Review.

P1 L39-40 It is inappropriate to define “activation gating device” as “a structural part of the NaV1.8 channel molecule” since it is obvious and uninformative. If the change in Zeff value suggests which part of the channel may be the binding site (residues set, transmembrane segments or domains), this should be indicated here or later. If there are no such grounds, explanations should not be given.

 P1 It is not clear why the first mention of the fact that “two positively charged guanidinium groups of the R1 and R3 arginine side chains are necessarily required for effective binding” and “the distances between the guanidinium groups (L42-44)... should fall within a certain range” (L48-50) is separated from the hypothesize “that if the distance between guanidinium groups responsible for ligand-receptor binding is less than the characteristic value of 9 Å, this peptide should not be able to modulate the NaV1.8 channel activation gating system” (L72-75). This can be stated more concisely in one paragraph.

P4. figure 2. The caption for Figures 2c and 2d should be rephrased since the current description is more appropriate as description in the text than as a caption. At least a title similar to "Normalized peak current-voltage functions ..." in Figure 1 should be given for data presented.

P5. figure 3. In connection with the statement in lines 507-515 and sentence “The Zeff value has decreased from 6.7 electron charge units in the control experiment to 4.7 electron charge units after the application of AcPRERRA-NH2 (Figure 2c), and from 6.6 to 4.8 after the application of Ac-PRARRA-NH2 131 (Figure 2d).” (P4 L129-131), it is not clear why it is possible to compare the single value Zeff in control condition with the values for five peptides, when, I suppose, Zeff after each peptide application should be compared with corresponding control Zeff value.

P6. L184. Could you explain what kind of the surrounding milieu does the ε value of 10 correspond to? Why was it chosen along with the value of 80?

P13 L414 I believe that investigation with living nerve cells might be considered rather as in vitro, not as in vivo experiment.

P13 L416-419 The statement: “Thus, the conclusions on  the possible analgesic effect of the applied agents are expected to be more reliable than  the results based on in vitro techniques and subject to further trial-and-error verification  in vivo”, looks very doubtful, since analgesic effect of peptides was not confirmed by none of the experiments presented here or previously.

The Discussion should be carefully revised to avoid duplications, in particular explanation the role of Nav1.8 in the nociception, goals, methodology and the obtained results. I also recommend to start it with a paragraphs P13 L420-433.

P14-15 L485-515 The paragraph 4.2. Patch-Clamp Method should be adjusted so that its content matches the style of the Materials and methods section.

Since both Nav1.8 and Nav1.9 are slow TTX-resistant channels expressed in DRG neurons how could you confirm that responses observed in electrophysiological experiments are belong only to Nav1.8 subtype?

Author Response

Reviewer 3

P1 L39-40 It is inappropriate to define “activation gating device” as “a structural part of the NaV1.8 channel molecule” since it is obvious and uninformative. If the change in Zeff value suggests which part of the channel may be the binding site (residues set, transmembrane segments or domains), this should be indicated here or later. If there are no such grounds, explanations should not be given.

The change in Zeff value does not indicate which part of the channel may be the binding site. We have corrected the term for the suggested molecular target and removed the explanation.

Lines 39, 45.

P1 It is not clear why the first mention of the fact that “two positively charged guanidinium groups of the R1 and R3 arginine side chains are necessarily required for effective binding” and “the distances between the guanidinium groups (L42-44)... should fall within a certain range” (L48-50) is separated from the hypothesize “that if the distance between guanidinium groups responsible for ligand-receptor binding is less than the characteristic value of 9 Å, this peptide should not be able to modulate the NaV1.8 channel activation gating system” (L72-75). This can be stated more concisely in one paragraph.

Done. Lines 50-61.

P4. figure 2. The caption for Figures 2c and 2d should be rephrased since the current description is more appropriate as description in the text than as a caption. At least a title similar to "Normalized peak current-voltage functions ..." in Figure 1 should be given for data presented.

Done. Lines 142-143.

P5. figure 3. In connection with the statement in lines 507-515 and sentence “The Zeff value has decreased from 6.7 electron charge units in the control experiment to 4.7 electron charge units after the application of AcPRERRA-NH2 (Figure 2c), and from 6.6 to 4.8 after the application of Ac-PRARRA-NH2 131 (Figure 2d).” (P4 L129-131), it is not clear why it is possible to compare the single value Zeff in control condition with the values for five peptides, when, I suppose, Zeff after each peptide application should be compared with corresponding control Zeff value.

Done. Lines 522-528

P6. L184. Could you explain what kind of the surrounding milieu does the ε value of 10 correspond to? Why was it chosen along with the value of 80?

Done. Lines 186-195

P13 L414 I believe that investigation with living nerve cells might be considered rather as in vitro, not as in vivo experiment.

Done. Lines 415-417

P13 L416-419 The statement: “Thus, the conclusions on the possible analgesic effect of the applied agents are expected to be more reliable than the results based on in vitro techniques and subject to further trial-and-error verification in vivo”, looks very doubtful, since analgesic effect of peptides was not confirmed by none of the experiments presented here or previously.

Done. Sentence deleted.

The Discussion should be carefully revised to avoid duplications, in particular explanation the role of Nav1.8 in the nociception, goals, methodology and the obtained results. I also recommend to start it with a paragraphs P13 L420-433.

Done.

P14-15 L485-515 The paragraph 4.2. Patch-Clamp Method should be adjusted so that its content matches the style of the Materials and methods section.

Done. Line 493

Since both Nav1.8 and Nav1.9 are slow TTX-resistant channels expressed in DRG neurons how could you confirm that responses observed in electrophysiological experiments are belong only to Nav1.8 subtype?

Done. Lines 581-591

Round 3

Reviewer 3 Report

Please, find comments in PDF file. All comments refer to the methods section.

Author Response

 Please see the attachment. All corrections are highlighted in green according to the comments. 
